# Current Strategies for Exosome Cargo Loading and Targeting Delivery

**DOI:** 10.3390/cells12101416

**Published:** 2023-05-17

**Authors:** Haifeng Zeng, Shaoshen Guo, Xuancheng Ren, Zhenkun Wu, Shuwen Liu, Xingang Yao

**Affiliations:** NMPA Key Laboratory for Research and Evaluation of Drug Metabolism, Guangdong Provincial Key Laboratory of New Drug Screening, School of Pharmaceutical Sciences, Southern Medical University, Guangzhou 510515, China; 3190405024@i.smu.edu.cn (H.Z.); 3190405022@i.smu.edu.cn (S.G.); 3190080016@i.smu.edu.cn (X.R.); 3190405026@i.smu.edu.cn (Z.W.)

**Keywords:** extracellular vesicles, exosomes, cargo loading, targeting delivery

## Abstract

Extracellular vesicles (EVs) such as ectosomes and exosomes have gained attention as promising natural carriers for drug delivery. Exosomes, which range from 30 to 100 nm in diameter, possess a lipid bilayer and are secreted by various cells. Due to their high biocompatibility, stability, and low immunogenicity, exosomes are favored as cargo carriers. The lipid bilayer membrane of exosomes also offers protection against cargo degradation, making them a desirable candidate for drug delivery. However, loading cargo into exosomes remains to be a challenge. Despite various strategies such as incubation, electroporation, sonication, extrusion, freeze–thaw cycling, and transfection that have been developed to facilitate cargo loading, inadequate efficiency still persists. This review offers an overview of current cargo delivery strategies using exosomes and summarizes recent approaches for loading small-molecule, nucleic acid, and protein drugs into exosomes. With insights from these studies, we provide ideas for more efficient and effective delivery of drug molecules by using exosomes.

## 1. Introduction

Extracellular vesicles (EVs) represent a heterogeneous group of membrane structures that undergo various biogenetic and secretory mechanisms to be released by cells. Exosomes, which are one of the subtypes of EVs, typically range from about 30 to 100 nm in diameter and are released into the extracellular space through a variety of cell types [1]. Exosomes are present in different body fluids, such as plasma [2], urine [3], milk [4], amniotic fluid [5], and ascites [6]. As scientists continue to gain deeper insights into the structure, biology, and function of exosomes, there is growing interest in exploring the potential for their clinical diagnostic and therapeutic applications across numerous diseases [7].

The formation and secretion of exosomes differ from typical microbubbles, as they involve complex cellular mechanisms. Endocytosis of plasma membrane by early endosomes and multivesicular bodies (MVBs) plays crucial roles in the transport and internalization of substances both inside and outside the cell. A consensus model postulates that the inward invagination of the cell membrane forms early endosomes during endocytosis, and further invagination of the early endosome surface generates late endosomes. Late endosomes, in turn, evolve into MVBs, allowing for the accumulation of intraluminal vesicles (ILVs) within MVBs [1]. Intracellular MVBs can be degraded in lysosomes after protein ISGylation or transported through interaction with actin and microtubule cytoskeletons to fuse with the plasma membrane and release exosomes [8]. Parental cells release exosomes through cytosolic transfer, and released exosomes can mediate cell-to-cell signaling locally or through in vivo blood transport to mediate long-distance intercellular communication. The process is remarkable since ILVs are present in endosomes during the maturation of late endosomes into MVBs, a process that involves specific sorting mechanisms for segregating cargo into specific regions of MVBs, with subsequent inward germination and division of small membrane vesicles containing isolated cytoplasmic sols, eventually resulting in the release of exosomes [9]. Therefore, this review mainly focuses on exosomes. If EVs are sometimes mentioned, it is because the cited article does not distinguish between EVs and exosomes.

## 2. Biological Functions of Exosomes

### 2.1. Exosomes Are Involved in Intercellular Signaling

After secretion from parental cells, exosomes can initiate downstream signaling cascades through cell surface protein interactions or receptor–ligand bindings with recipient cells. Additionally, they can release their contents through endocytosis after fusion with the plasma membrane of recipient cells, thus, allowing for intercellular signaling [10,11]. Exosomes exhibit functionality, targeting, and accumulation of specific cellular components driven by various mechanisms, highlighting their crucial role in regulating intercellular communication (Figure 1A). Ferreira et al. demonstrated that the membrane protein LAMP2A could load hypoxia-inducible transcription factor 1α (HIF1A) into exosomes, which could then transmit hypoxic signals to normoxic cells, thereby, activating neovascularization in vivo [12]. In addition, different types of EVs are able to carry organelles such as mitochondria and ribosomes for intercellular crosstalk or parental cell self-regulation [13,14]. Neural stem cells (NSCs) are capable of delivering mitochondria to target cells via EVs to rescue mitochondrial dysfunction [15]. Researchers have constructed exosomes loaded with cell-specific cargoes such as proteins, lipids, and nucleic acids, thereby, creating new intercellular communication mechanisms with other target cells and eliciting specific, signal-mediated responses.

### 2.2. Exosomes Are Involved in Antigen Presentation and Regulation of Immune Responses

The role of exosomes in the immune response has been extensively investigated (Figure 1B). Notably, viral transmembrane domains (TMs) can efficiently load antigens onto the surface of exosomes, which are taken up by B lymphocytes through antigen presentation and induce enhanced humoral and cell-mediated immune responses [16]. Importantly, administration of Wharton’s jelly-derived mesenchymal stem cells (WJMSCs) into patients with acute graft versus host disease (aGVHD) has been reported to result in the production of a large number of immunosuppressive exosomes [17]. These exosomes were found to be highly enriched with PD-L1, and treatment with these exosomes significantly suppressed T cell receptor-mediated activation of T cells in a dose-dependent manner. Therefore, provision of such cell-free therapies based on WJMSC-derived exosomes may represent a potential strategy for the treatment of aGVHD. Additionally, exosomes derived from immature dendritic cells (imDC) have low immunogenicity due to reduced expression of MHC-I, CD40, and other surface markers, and therefore, they are highly suitable for drug delivery and in vivo treatment applications [18]. The cargoes carried by exosomes, for example, DNA and microRNA, can modulate both the innate and adaptive immune responses. Intriguingly, cell-intrinsic N6-methyladenosine reader protein (YTHDF1) in tumors can disrupt tumor antigens and impair immune surveillance, whereas engineered exosomes can be utilized to deliver CRISPR/Cas9 in order to specifically target tumor YTHDF1, thereby, restoring tumor immune surveillance [19]. Furthermore, a recent study by Bai et al., demonstrated that placenta-derived exosomes (pEXO) could interact with and modulate the phenotype of monocytes in the maternal bloodstream during pregnancy, reducing the immune response and establishing maternal immune tolerance towards the fetus [20]. Collectively, these findings highlight the critical and diverse roles of exosomes in the immune system and demonstrate their potential as an anti-inflammatory and immune-modulating therapeutic strategy for various diseases.

### 2.3. Exosomes Are Involved in Tumor Development

The tumor microenvironment (TME) comprises the cellular environment that tumors depend on, which includes endothelial cells, fibroblasts, immune cells, and extracellular matrix. Hypoxia is an important feature of the TME that regulates the biogenesis of exosomes in tumor cells [21]. Exosomes carrying various cargoes, such as proteins and nucleic acids, play crucial roles in mediating signaling between cancer and stromal cells, in their cellular environment [18]. In pancreatic cancer cells, hypoxia induces the upregulation of circular PDK1, which activates the host gene PDK1. Exosomes with high expression of circular PDK1 activate c-Myc by regulating the miR-628-3p/BPTF axis and circular PDK1-BIN1 axis, thus, promoting tumor growth, metastasis, and glycolysis [22]. Tumor cell-derived exosomes carrying angiogenesis-stimulating factors, such as VEGF, FGF, and IL-8, can induce tumor angiogenesis [23] (Figure 1C). Moreover, exosomes derived from colorectal cancer cells containing circular TUBGCP4 can promote angiogenesis and tumor metastasis by activating the Akt signaling pathway in vascular endothelial cells [24]. Tumor-derived exosomes also mediate EMT, anti-apoptotic pathways, and drug efflux, which increase drug resistance in tumor cells [23]. As the mechanisms underlying the role of exosomes in tumor development become increasingly clear, targeted tumor cell therapy through engineered exosomes is gaining momentum and legitimacy as a viable therapeutic strategy.

### 2.4. The Role of Exosomes for Wound Healing

Wound healing involves the repair of damaged epidermis or dermis of the skin, for which multifunctional hydrogel dressings and skin substitutes have been extensively studied [25,26]. However, these approaches have limitations such as slow wound healing, immune rejection, and low mechanical strength. In recent years, stem cells have gained favor due to their self-renewal and differentiation abilities [27]. Exosomes derived from mesenchymal stromal cells (MSCs) have been reported to hold great potential for wound healing [28]. Exosomes provide a cell-free therapy option and possess the advantages of easy administration, low immunogenicity, and the ability to control inflammation, promote cell proliferation, and angiogenesis (Figure 1D). A targeted engineered exosome (i.e., SGM-miR146a-Exo@SFP) has been developed for diabetic wound healing, demonstrating the potential of exosomes [29]. In terms of wound healing, exosomes derived from stem cells regulate the inflammatory response by controlling the polarization of inflammatory cells and macrophages, promoting epithelial regeneration by activating fibroblasts and keratinocytes, stimulating the release of angiogenic factors and endothelial cell activity to promote angiogenesis, and improving tissue remodeling by regulating the ratio of collagen and myofibroblast differentiation.

### 2.5. The Role of Exosomes in Disease Diagnosis

Exosomes, which are present in various body fluids, carry different types of contents such as proteins, lipids, and RNA, which exhibit diverse degrees of parental cell specificity (Figure 1E). Given the fact that exosomes can exist stably in bodily fluids while carrying specific biomolecules, liquid biopsy techniques based on exosomes have been extensively researched for diagnosing various diseases [30,31]. In the context of asthma pathology, exosomes containing inflammatory miRNAs have been released into serum, with the levels of miR-21-5p, miR-126-3p, and miR-146a-5p in serum exosomes serving as potential indicators of the severity of asthma [32]. In a study by Xiang et al., the levels of carbamoyl phosphate synthase 1 (CPS1) in serum exosomes from healthy controls, those with acute hepatitis E (AHE), and those with hepatitis E virus-associated acute liver failure (HEV-ALF) were compared, and it was found that CPS1 levels in serum exosomes of HEV-ALF patients were significantly higher compared to the other two groups. It was suggested that serum exosome-derived CPS1 levels might serve as viable biomarkers for accurately diagnosing and predicting the onset of acute liver failure, as well as its fatality rates [33]. Crucially, exosomes also hold immense potential in the domain of cancer diagnosis, as exemplified by a study conducted by Zhao et al. They developed an automated centrifugal microfluidic disc system and incorporated functionalized membranes (Exo-CMDS) to isolate and enrich exosomes, followed by a novel aptamer fluorescence system (Exo-AFS) for detecting exosomal protein biomarkers indicating lung cancer, such as PD-L1, CA125, CD63, CEA, and EpCAM. This cutting-edge diagnostic instrument is poised to become one of the primary tools for lung cancer screening [34]. Further, miR-9-3p in exosomes has shown potential for diagnosing hepatocellular carcinoma, while lncUEGC1 can be leveraged to screen early gastric cancer. Exosomes containing miR-15b-3p, lnc-GNAQ-6, linc-00152, and lncHOTTIP have also been identified as being indicative of gastric cancer [35,36]. In summary, given their multifaceted involvement in cancer, exosomes represent ideal diagnostic biomarkers and therapeutic targets that can be exploited in a variety of medical contexts.

### 2.6. The Role of Exosomes in Drug Delivery

Exosomes possess unique characteristics such as stability, low immunogenicity, and high biocompatibility which make them ideal drug delivery platforms [37,38]. These features enable exosomes to effectively enter target cells, avoiding recognition and elimination by the immune system, and therefore, deliver foreign protein and nucleic acid drugs to target cells. For instance, endothelium-derived exosomes exhibit excellent skeletal targeting and anti-osteoclast activity (Figure 1F). Moreover, loading miRNA into exosomes avoids immune stimulation and provides a suitable therapeutic tool for fracture repair [39]. Exosomes can also improve drug utilization by increasing drug solubility and enabling co-delivery of multiple drugs due to their lipid bilayer structure with lipophilic and hydrophilic characteristics [40]. They also have the ability to enter the blood-brain barrier and effectively deliver drugs to intracerebral regions for the treatment of CNS disorders using modified or unmodified exosomes [41]. Furthermore, the lipid bilayer structure of exosomes can be modified to enhance their targeting specificity [42]. Therefore, in this manuscript, we mainly review recent strategies in exosome drug delivery.

## 3. Advantages of Exosomes as Drug Carriers

Compared to other nanocarriers such as liposomes and polymeric nanoparticles, exosomes are natural carriers for intercellular transfer of genetic material, as well as mediators of gene expression in recipient cells [43]. They are present in various body fluids and possess many advantages over traditional synthetic delivery vectors, including better biocompatibility, stronger stability, lower immunogenicity, high stability in blood, and direct drug delivery to cells. The protein composition and lipid content of exosomes confer targeting abilities towards specific organs [44]. Exosomes can be transported between cells, enabling the exchange of substances and information and altering the functional state of recipient cells by loading exogenous drugs, such as small-molecule, transmembrane protein, or nucleic acid drugs [37]. Exosomes have shown promise as drug delivery vehicles in therapeutic research. For instance, loading patient-specific neoantigens into dendritic cell-derived exosomes for cancer immunotherapy has shown promising results, with exosome-based nano vaccines inhibiting tumor growth and promoting tumor cell uptake more effectively than liposomes [45]. Furthermore, exosomes isolated from lens epithelial cells and loaded with the drug doxorubicin (DOX) have demonstrated the potential to prevent posterior capsule opacification complications after cataract surgery by delivering the drug specifically to target cells through the targeting ability of exosomes [46]. Highly metastatic colorectal cancer cell-derived exosomes can deliver miR-181a-5p to hepatic stellate cells, promoting tumor liver metastasis through the activation of hepatic stellate cells or tumor microenvironment remodeling [47]. Additionally, modification of bone marrow mesenchymal stem cell-derived exosomes with a specific short peptide for heme oxygenase-1 (HSSP) and loading temozolomide or siRNA into these exosomes have demonstrated excellent tumor cell targeting capability, due to the overexpression of heme oxygenase-1 in glioblastoma [48]. Based on these early clinical data and numerous preclinical studies, exosomes are being developed as therapeutic agents.

## 4. Strategies for Loading of Small-Molecule Drugs

Exosomes have gained significant attention in recent years as a potential delivery system owing to their functional attributes. These vesicles offer several advantages, such as low immunogenicity, high biocompatibility, and specificity for target tissues [37]. The cargo inside exosomes remains stable and is not susceptible to degradation from enzymes or other agents due to their lipid bilayer structure. Additionally, exosomes possess remarkable properties that make them an ideal delivery system for small-molecule drugs. For instance, they can load natural cargoes, such as miRNA, siRNA, DNA, and proteins, and can deliver them to target sites with precision [49]. Qu et al., demonstrated that unmodified blood exosomes effectively targeted the brain and were capable of delivering dopamine as a treatment for Parkinson’s disease [50]. Likewise, Zhang et al., illustrated that mesenchymal stromal cell-derived exosomes carrying miR-101 could suppress osteosarcoma cell invasion and metastasis by downregulating B-cell lymphoma 6 (BCL6) expression [51]. The use of exosomes loaded with small-molecule drugs has shown promising therapeutic outcomes, and this delivery system can mitigate the adverse effects of drugs while preserving their pharmacological activities [52,53]. Nevertheless, a significant drawback is the low efficiency of exosome drug loading, and more research is required to enhance the efficiency of such loading methods (Figure 2).

### 4.1. Incubation for Loading of Small-Molecule Drugs

The incubation method, as its name suggests, relies on the co-incubation of exosomes with small molecules over a certain period. Generally, small molecules can be loaded into exosomes during this process. This approach is simple to perform and does not affect the integrity of the exosomes. The loading efficiency with this method depends on the polarity of the small molecule. Low-to-medium molecular weight lipophilic small molecules, such as catalase, are more readily loaded into exosomes via incubation. For instance, Wang et al., modified exosomes with RGD peptide, and incubated them with DOX to achieve drug loading. The ^131^I-labeled exosomes could target tumor cells due to the presence of the targeting peptide RGD, thus, demonstrating dual antitumor strategies of internal irradiation and chemotherapy [54]. Faruque et al., consecutively modified the functional ligand RCD and magnetic nanoparticles on exosomes derived from human pancreatic ductal carcinoma cells, followed by incubation with paclitaxel (PTX) to obtain Exo-PTX. It was found that Exo-PTX exhibited higher cytotoxicity and significantly targeted antitumor effects [55]. Yang et al., mixed linezolid with exosomes derived from macrophages and incubated them. The lipid content of the exosomes did not change significantly after incubation, and the exosomes demonstrated potent killing of methicillin-resistant *Staphylococcus aureus* (MRSA) [56]. However, the incubation method has a disadvantage of relatively low loading efficiency, which limits its application as a standalone method. Therefore, the incubation method is usually combined with other methods to improve the loading of small-molecule drugs.

### 4.2. Electroporation for Loading of Small-Molecule Drugs

Electroporation involves the loading of small-molecule drugs into exosomes through the application of an external electric field just beyond the exosome membrane, using short high-voltage pulses to overcome the barrier of the membrane. Under appropriate conditions, the lipid layer instantaneously ruptures, creating a transient state of membrane permeability, allowing small-molecule drugs to diffuse through the exosome membrane. Typically, researchers incubate electroporated exosomes for a period to restore membrane integrity [57]. Several studies have demonstrated the efficacy of electroporation for loading different molecules into exosomes. For instance, Zhu et al., successfully loaded DOX into exosomes derived from lens epithelial cells through electroporation and immobilized the loaded exosomes on the surface of an artificial lens, leading to effective uptake by lens epithelial cells and significant anti-proliferative effects [46]. Similarly, Wu et al., loaded HAL into M2 macrophage-derived exosomes using electroporation, which exhibited potent anti-inflammatory effects and ultimately led to remission of atherosclerosis [57]. Electroporation is a simple, efficient method for loading drugs into exosomes that maintains the original drug properties and offers superior loading efficiency compared with other methods. However, loading efficiency varies with electroporation conditions. Lennaárd et al., systematically evaluated various electroporation conditions, including total number of EVs, drug to vesicle ratio, electroporation buffer solution, pulse capacitance, and field strength, and demonstrated that loading efficiency could be significantly improved under appropriate electroporation conditions [58]. Similarly, Jia et al., found that a concentration ratio of curcumin to exosomes of 3:1 and a concentration ratio of superparamagnetic iron oxide nanoparticles to exosomes of 1:10, with electroporation at 400 V, 150 μF, and discharge time of 1 ms, produced optimal results [59].

### 4.3. Sonication for Loading of Small-Molecule Drugs

Small-molecule drugs can be loaded into exosomes through sonication, whereby, an ultrasound probe applies mechanical shear force to deform exosome membrane and allow drugs to enter the exosomes [60]. This loading method has yielded positive results, with researchers reporting good drug loading efficiency and enhanced antitumor effects. For instance, Wang et al., mixed PTX and M1 macrophage-derived exosomes in a certain ratio and subjected them to sonication, followed by incubation of the mixture at 37 °C for 1 h to recover the exosome membrane [61]. Yerneni et al., loaded albumin and curcumin sequentially into exosomes through sonication, resulting in carrier exosomes (CA-EVs) with high stability and anti-inflammatory effects [40]. Du et al., loaded ferroptosis inducer and photosensitizer into exosomes through sonication and engineered the exosomes to evade phagocytosis by the mononuclear phagocyte system [62]. Despite its advantages of high drug loading efficiency and continuous drug release, sonication can have adverse effects on exosome structure, inducing changes to the spherical shape and hydrophobic drug loading efficiency, as well as causing exosome aggregation [63]. Nonetheless, sonication remains a popular method for developing novel exosome drug delivery systems due to its strong potential for drug delivery optimization.

### 4.4. Extrusion for Loading of Small-Molecule Drugs

Extrusion-based techniques are commonly employed for loading small-molecule drugs into isolated exosomes. In this method, a mixture of the drug and isolated exosomes is passed through a lipid extruder equipped with porous membranes of various pore sizes (100–400 nm) at controlled temperatures. This process disrupts the exosome membrane, enabling efficient loading of the drug, and the strategy can also be extended to membrane hybridization of exosomes [64]. Le Saux et al., demonstrated that extrusion of exosomes through a 50 nm polycarbonate membrane at room temperature reduced exosome size, while preserving their spherical structure, internalization capacity, and protein content [60]. Zhang et al., used this approach to load PTX into exosomes isolated from human umbilical cord-derived mesenchymal stem cells (huc-MSCs). They extruded the PTX solution mixed with exosomes through a lipid extruder with a pore size of 100 nm, and achieved efficient PTX encapsulation into exosomes without any significant alteration in the lipid content of the loaded exosome membranes. The loading efficiency for PTX was recorded to be 14.23 ± 0.25%, higher than the efficiency obtained using the conventional freeze–thaw treatment (6.87 ± 0.4%). The drug-loaded exosomes were successful in inducing differentiation of neural stem cells into neurons [65]. Moreover, this method has been utilized to fabricate exosome-mimicking nanovesicles (NVs) for drug delivery. Zhang et al., employed a sequential extrusion method to isolate NVs from induced pluripotent stem cells differentiated from endothelial cells. Then, they passed a dapagliflozin (DA) solution mixed with NVs through 400 and 200 nm polycarbonate membranes repeatedly (10–20 times) using a mini extruder for drug loading. Under optimal conditions, the exosome mimics showed a loading efficiency of 45% (DA/NV (mg/mg) = 0.09) and were capable of targeting endothelial cells for drug delivery in vivo. They were also effective in promoting diabetic wound healing via the HIF-1α/VEGFA pathway [66].

### 4.5. Chimeric Exosome Method for Loading of Small-Molecule Drugs

Exosomes have the characteristics of endogenous nanocarriers, however, exosomes have a relatively small capacity, low loading efficiency, and tend to aggregate. In contrast, liposomes are capable of loading large-molecule drugs with greater flexibility in surface modification and also have good pharmacokinetic characteristics, but liposomes have relatively high cytotoxicity [67,68,69]. Based on these two properties, fusing exosomes with liposomes to form exosome-liposome hybrids can significantly increase their loading capacity and the half-life of exosomes in plasma, while also alleviating the cytotoxicity problems of liposomes [70]. Li et al., incorporated triptolide (TP) into tumor-targeting peptide RGD-modified liposomes and mixed it with tumor-derived exosomes expressing CD47, then performed vortex flow and sonication, vacuum vortexed for 15 min to remove the organic phase, and then hybrid nanoparticles containing TP were obtained by extrusion; miR497 was loaded into the above hybrid nanoparticles by subsequent experiments and used for the treatment of cisplatin-resistant ovarian cancer [71]. Lv et al., fused exosomes with heat-sensitive liposomes loaded with docetaxel (DTX) via a freeze–thaw procedure to obtain genetically engineered exosomal heat-sensitive liposome hybrid nanoparticles (gETL NPS), while DTX was loaded into them. They found that gETL NPS could accumulate in tumors after intravenous administration and could release DTX under the low temperature conditions of hot intraperitoneal chemotherapy (HIPEC) [72]. Sun et al., loaded clodronate (CLD) into liposomes, and then mixed it with fibroblast-derived exosomes via vortexing and sonication, followed by extrusion, ten times, through 400 nm and 200 nm polycarbonate membranes, sequentially, to achieve hybridization (EL-CLD). They found that EL-CLD was effective in avoiding liver uptake and targeting lung fibrotic tissues during the second dose within 48 h after the first dose. Based on the excellent tissue targeting properties of EL-CLD, the researchers mixed nintedanib (NIN) with EL-CLD in chloroform to load NIN into EL-CLD, which greatly enhanced the therapeutic effect of NIN and effectively delayed the progression of pulmonary fibrosis [64]. However, comprehensive evaluations of exosome-liposome hybrids are lacking, and their adverse effects on recipient cells are not yet clear [73].

### 4.6. Endogenous Loading for Small-Molecule Drugs

Endogenous loading is an engineered method for drug loading that utilizes exosome donor cells. In this approach, small-molecule drugs are co-incubated with donor cells or treated with other loading strategies such that they can be absorbed by the cells through the lipid bilayer and encapsulated in exosomes. The resulting exosomes contain the desired small-molecule drugs. This method is simple and convenient, but due to the sensitivity of cells to small-molecule drugs, it is generally only suitable for drugs with low cytotoxicity [74]. Researchers have explored several strategies to improve drug loading efficiency. For example, Sancho-Albero et al., reported an increase in stability and intracellular internalization of polyethylene glycol-coated gold nanoparticles (PEG-HGNs) when added at a subcytotoxic dose to murine melanoma cells. This led to a loading efficiency of 50%, higher than other methods for loading into exosomes such as electroporation, sonication, and heat shock [74]. Another example is the work of Ma et al., who incorporated MCC950, an NLRP3-inflammasome inhibitor, into platelets. By activating the platelets, they were able to obtain EVs (MCC950-PEVs). MCC950-PEVs were found to selectively target macrophages in atherosclerotic plaques and significantly reduce plaque formation in vivo [75]. These advances in endogenous loading and other drug loading strategies are promising for targeted drug delivery and hold potential for future medical applications.

## 5. Loading Strategies for Nucleic Acid Drugs

Nucleic acid drugs are natural biopolymers that possess high hydrophilicity and negative charges. However, nucleic acid drugs alone are unable to enter cells due to various nucleases and hepatic and renal clearance, which leads to their degradation. The instability and degradability of nucleic acid drugs greatly affect their delivery, making them heavily reliant on the protection of drug delivery systems [76]. Consequently, the concept of loading nucleic acid drugs into exosomes for drug delivery has emerged. Compared to lipid nanoparticles, exosomes are less cytotoxic and exhibit higher cargo expression, making them effective drug delivery vehicles that protect nucleic acid drugs from degradation and enhance their stability [77]. Exosomes are characterized by their natural capacity to deliver miRNAs and siRNAs, which bind to mRNAs in target cells, resulting in the degradation of mRNA or inhibition of protein response processes. This leads to the silencing of target genes through a process known as RNA interference [78]. Based on this process, researchers have shifted their focus to small RNA-based therapies.

### 5.1. Incubation for Loading of Nucleic Acid Drugs

Similar to small-molecule drugs, the aforementioned method is also applicable to loading nucleic acid drugs. Gong et al. demonstrated successful loading of macrophage-derived exosomes with miRNA by incubating cholesterol-modified miR-159 with exosomes, while shaking at 500 rpm. Notably, the loading efficiency of cholesterol-modified miR-159 into exosomes was found to be higher than that of miR-159 alone [79]. RNA modified by using cholesterol rendered asymmetric RNA oligonucleotides with a hydrophobic portion, thereby increasing their stability due to the presence of cholesterol. These modifications facilitate rapid binding to the cell membrane, promote cellular internalization, and induce effective gene silencing [80,81]. Haraszti et al. coupled cholesterol to siRNA through both of the triethyl glycol (TEG) and 2-aminobutyl-1-3-propanediol (C7) junctions, and incubated it with exosomes derived from mesenchymal stem cells obtained from umbilical cord Wharton’s jelly. In doing so, they observed a loading efficiency of 43% when the ratio of siRNA to exosomes was 6000. With increasing siRNA to exosome ratios, the loading efficiency gradually decreased, eventually reaching a plateau efficiency of 18%. Additionally, they determined that the optimal loading amount of each exosome was 3000 siRNA copies [80]. Hade et al. incubated cell-penetrating peptide YARA-coupled miR-21-5p with MSCs-derived exosomes at room temperature for varying times, and observed a slight increase in the diameter of the loaded exosomes, which was accompanied by a decrease in zeta potential. The loading of exosomes gradually increased with increasing incubation time and peaked at 8 h of incubation, where each exosome contained 1600 copies of miRNA. The YARA modification of miRNAs significantly increased the loading effect of exosomes compared to naked miRNAs, while also enhancing proliferation, migration, and invasion of fibroblasts. This study, thus, provides a novel delivery strategy of miRNAs for wound healing [82].

### 5.2. Electroporation for Loading of Nucleic Acid Drugs

Electroporation is a widely used method for loading nucleic acid drugs and has been applied in various studies. Rong et al., immobilized the target therapeutic peptide CAQK onto the exosome membrane, subsequently mixed the exosomes with siRNA, and performed electroporation under specific conditions. Furthermore, induced neural stem cell-derived exosomes exhibited anti-inflammatory and neuro reparative effects, and the co-delivery of siRNA enhanced the repair effect on spinal cord injury [83]. It has been shown that aggregation or precipitation of siRNA occurs when loading siRNA by electroporation [84]. However, aggregation or precipitation of siRNA might occur during electroporation. Liang et al., discovered that aggregation and encapsulation of nucleic acids existed simultaneously during electroporation loading of miR-21 inhibitor oligonucleotides and 5-fluorouracil. The highest drug loading efficiency was achieved at electroporation conditions of 10 ms and 1000 V [85]. Kim et al., used T7 peptide-modified exosomes to deliver miRNA-21 to intracranial tumor cells. The T7-exo was loaded with the drug by electroporation at 400 V, with a loading efficiency of 1.68 ± 0.23% [86]. Zhao et al., employed cellular nanoporation (CNP) for miRNA-484 loading into RGD-modified exosomes, and found the highest loading efficiency at 700 V [87]. CNP has been shown to be more effective than electroporation for loading endogenous RNA into exosomes [88].

### 5.3. Sonication for Loading Nucleic Acid Drugs

Sonication has proven to be preferable to electroporation for loading exosomes with nucleic acids and can efficiently load various types of small-molecule and nucleic acid drugs. In one study, Tao et al., isolated exosomes from milk, dispersed them with PBS, and added Bcl-2 siRNA at a mass ratio of exosome to siRNA of 5:1. These exosomes were capable of targeting cancer cells by upregulating apoptotic genes and downregulating metastasis-related genes-inducing apoptosis, suppressing tumor migration and invasion [89]. Xiang et al., mixed milk-derived exosomes with siRNA-Keap1 (siKeap1) in equal proportions (mass/mass), and subjected them to sonication. The resulting loaded exosomes (mEXOs-siKeap1) had a slightly increased diameter, and a loading efficiency of 24%. The researchers observed that mEXOs-siKeap1 was able to reduce Keap1 protein levels in methylglyoxal-treated human umbilical vein endothelial cells, and promoted wound healing in diabetic wounds by facilitating collagen formation and neovascularization in vivo [4].

### 5.4. Transfection for Loading Nucleic Acid Drugs

The transfer of specific plasmids into exosome donor cells for expressing a desired nucleic acid drug, which is subsequently encapsulated and secreted from exosomes, involves the use of transfection reagents. Born et al., employed this method by mixing pCMV-HOTAIR plasmid with P3000 reagent, followed by the addition of liposome solution. Then, the mixed solution was added to human bone marrow-derived mesenchymal stem/stromal cells (BDMSCs), and exosomes containing HOTAIR were obtained after multiple rounds of centrifugation. The high amounts of HOTAIR in exosomes were effective in promoting angiogenesis and wound healing [90]. Yu et al., produced exosomes containing circDYM (RVG-circDYM-EVs) through differential centrifugation after co-transfecting circDYM-GFP lentivirus and RVG-Lamp2b plasmid with HEK-293T cells using Lipofectamine 2000. Administration of RVG-circDYM-EVs facilitated a marked increase in the level of circDYM in the hippocampal tissue of the brains of chronic unpredictable stress (CUS) mice, while mitigating CUS-induced hippocampal inflammatory response and astrocyte dysfunction [91]. In addition to transfection of exosome donor cells, exosomes containing nucleic acid drugs can also be obtained through direct transfection of exosomes. de Abreu et al., achieved this by mixing small EVs (sEVs) with Exo-Fect (commercial transfection kit) followed by incubation at 37 °C for 10 min. They compared the loading efficiency of miRNAs under different methods and found Exo-Fect-based transfection to be the most efficient loading strategy. Exo-Fect interferes with the membrane of sEVs while protecting miRNAs from enzymatic degradation, enhancing intracellular transport and delivery of cargo [92].This approach has high loading efficiency and molecular stability and does not require specific experimental equipment. However, certain conventional transfection reagents are toxic and may pose safety issues [93]. Furthermore, recent work has revealed that transfection reagents may contaminate EVs during transfection, and conventional differential centrifugation cannot separate transfection complexes from EVs. This can result in masking of the delivery capability of EVs and may also affect the transfer of cargo within EVs to recipient cells [94,95].

## 6. Loading Strategies for Protein Drugs

With the ongoing advancements in genetic engineering, protein drugs have become increasingly popular due to their high efficacy, specificity, and low toxicity. However, the immunogenicity of protein drugs often leads to various immune responses and allergic reactions in humans [96]. In addition to this, their stability can be compromised, necessitating different dosage forms and chemical modifications to maintain their biological activity [97]. Thus, delivering protein drugs safely and effectively has emerged as a major challenge in current research. Exosomes can be a promising option in this regard, as protein drugs can be encapsulated into them by either genetic engineering or exogenous loading methods for efficient drug delivery [98,99]. Haney et al., recently proposed a novel exosome-based delivery system for the treatment of Parkinson’s disease (PD) by loading catalase onto exosomes using various techniques (room temperature incubation, saponin permeation, freeze–thaw cycles, ultrasound, or extrusion) and evaluated their efficacy in vitro [100].

### 6.1. Incubation for Loading Protein Drugs

Proteins can be directly encapsulated into exosomes by co-incubation with exosomes. Haney et al. used four different methods to load catalase into macrophage-derived exosomes and evaluated these methods, in which they mixed exosomes with catalase solution after incubation for 18 h at room temperature; they obtained a loading efficiency of 4.9 ± 0.5% for this method. In addition, they found that their loading efficiency was considerably higher (18.5 ± 1.3%) when 0.2% saponin was added [100]. Fuhrmann et al., isolated exosomes from human mesenchymal stem cells (hMSCs), mixed them with β-glucuronidase and saponin, and incubated them to load β-glucuronidase into the exosomes, and found that the loading efficiency was comparable to that of liposomal carriers and that saponin did not disrupt the activity of β-glucuronidase. The first EV-based hydrogel delivery system was developed by incorporating the loaded exosomes into PVA hydrogels to reduce the release of β-glucuronidase and to maintain the stability of the enzyme under long-term incubation [101].

### 6.2. Electroporation for Loading Protein Drugs

In addition to loading small-molecule and nucleic acid drugs, electroporation can be employed to directly load protein drugs into exosomes. Although incubation is a possible method for introducing protein drugs into exosomes, electroporation appears to have higher loading efficiency. Rodríguez-Morales et al., successfully obtained exosomes containing insulin by mixing exosomes with insulin (in a volume of 400 μL) and subjecting them to electroporation, followed by incubation at 37 °C for 1 h. They found that the highest loading efficiency was achieved when the electroporation conditions were 200 V and 50 μF, and that the loading efficiency of exosomes from different cell sources varied under the same electroporation conditions. Notably, the efficiency of this electroporation method was significantly higher than that of exosome loading using the room temperature incubation treatment [102]. Lu et al., mixed recombinant Yap1 protein and platelet-derived exosomes in the electroporation buffer followed by their electroporation using a Gene Pulser II electroporation system. rPLT-Exo-Yap1 could suppress oxidative stress-induced senescence-related phenotypes by inhibiting the Ikβα and NF-κB axes. Furthermore, an rPLT-Exo-Yap1-functionalized GelMA hydrogel was capable of promoting the stemness of tendon stem/progenitor cells (TSPCs) and functional recovery of endogenous TSPCs, thereby, supporting tendon regeneration [103]. Wan et al., added Cas9 ribonucleoprotein complex (RNP) in electroporation buffer and subjected the mixture to electroporation to form exosome complexes without significant changes in the size and morphology of the exosomes. A Western blot analysis showed that the loading efficiency was significantly higher than that of other methods. The administered exosome complexes could be effectively targeted to the liver and used for the treatment of liver diseases [98].

### 6.3. Sonication for Loading of Protein Drugs

For catalase loading, sonication achieves high enzyme loading efficiency; the reorganization of exosomes under sonication allows catalase to pass through the lipid bilayer into the exosomes, while the structure of catalase remains stable, resulting in its continuous release from the exosomes [100]. Haney et al., similarly utilized sonication to load exosomes with TPP1, a lysosomal enzyme with solubility. Although the exoxome sizes became larger, their morphology did not significantly change. The amount of TPP1 in exosomes under sonication was higher than under saponin treatment at the same level. Furthermore, loading TPP1 into exosomes significantly improved its stability and allowed it to be delivered effectively to the central nervous system, ultimately improving neurological function [63]. Finally, Zhuang et al., sonicated a mixture of RNP and exosomes in an ice bath, and the loading efficiency was determined by Western blot and BCA. Then, TDNs were modified on the exosome surface by cholesterol anchoring to target specific cells [104].

### 6.4. Freeze–Thaw for Loading Protein Drugs

The freeze–thaw cycling method represents a simple and effective strategy to load nucleic acid drugs directly into exosomes. This process involves the repeated freezing of exosomes at low temperatures, followed by thawing at room temperature, which induces the exosomal plasma membrane to rupture and repair multiple times. In the continuous rupture and repair process, the drug enters into the exosome, thereby, achieving exosome drug loading. The freeze–thaw cycle is a mild procedure that does not damage the membrane structure of the exosome, making it a viable option for mass production, since the exosomes remain stable at low temperatures. Moreover, the biological activity of the original exosomes is also maintained during this process, enhancing its appeal as a drug delivery platform [105]. Catalase is able to be loaded into exosomes by using the freeze–thaw cycle, which is less efficient than sonication and extrusion, while the exosomes treated by using the freeze–thaw cycle have larger particles, probably due to aggregation [100]. Hajipour et al. used the freeze–thaw cycle to load human chorionic gonadotropin (hCG) into exosomes. The encapsulation rate and loading capacity of exosomes under freeze–thaw cycles were calculated to be 14.02 ± 5.46% and 245.06 ± 95.66 IU/mg, respectively. These values were inferior to those achieved by sonication (500 V, 20% power, six cycles with 4 s pulse/2 s pause, frequency of 2 kHz), which resulted in respective values of 40.55 ± 4.21% and 710.05 ± 73.74 IU/mg under the same conditions [106]. For exosome loading of RNP, Zhuang et al. also employed a freeze–thaw cycling approach. The RNP solution was added to exosomes and incubated for 30 min at room temperature, immediately frozen with liquid nitrogen, and then thawed at room temperature. This process was repeated three times. Compared to ultrasonication under the same conditions, the freeze–thaw cycle achieved a loading efficiency of 37.62%, making it well suited for subsequent targeted delivery of exosomes [104].

### 6.5. Transfection for Loading of Protein Drugs

Drug-loaded exosomes are obtained through transfection reagents used to express or load proteins or peptides onto the surface of exosomes. Zou et al., transfected 293T cells with an interferon-induced transmembrane protein 3 (IFITM3) expression vector using polyethyleneimine (PEI) transfection reagent. Subsequently, the cells were cultured at 37 °C in 5% humidified atmosphere for 48 h, followed by isolation of IFITM3-Exos through differential centrifugation. Flow cytometry revealed that 55.5% of the CD81-positive exosomes expressed IFITM3. Administration of IFITM3-Exos via tail vein showed that it crossed the placenta into the mouse fetus, effectively inhibiting Zika virus infection in pregnant mice and fetuses without causing any harm [107]. In another study, Chen et al., loaded Cas9 protein into exosomes by transfecting HeLa (a human cervical cancer cell line) and HuH7 (a human liver cancer cell line) cells with human papillomavirus-specific CRISPR/Cas9 expression plasmids and hepatitis B virus-specific CRISPR/Cas9 expression plasmids for 48 h, respectively. This led to demonstrating that gRNA and Cas9 protein could be carried independently by endogenous exosomes [108]. Majeau et al., mixed RNP with CM (the protein transfectant Lipofectamine CRISPRMAX) and added purified serum exosomes, leading to the incorporation of RNP into the exosomes in the presence of CM. These exosomes were capable of efficiently delivering the active SpCas9 protein to muscle fibers, generating up to 10% gene modification from double cleavage of the CRISPR system in different mouse models, providing an excellent delivery tool for the treatment of Duchenne muscular dystrophy (DMD). Furthermore, it was found that exosomes isolated under different purification methods had varying incorporation effects on RNP, with higher uptake of RNP by exosomes isolated by size-exclusion chromatography (SEC) (96%) than by exosomes isolated by ultracentrifugation (UC) (39%) [109].

### 6.6. Supplemental Loading Strategies for Protein Drugs

Haney et al., utilized extrusion to incorporate catalase into exosomes. The catalase-exosome mixture was passed through an extruder with a pore size of 200 nm, and the resulting ExoCAT was purified by gel filtration chromatography. The loading efficiency of catalase in ExoCAT was determined to be 22.2 ± 3.1%, and catalase remained stable within the exosomes. Using atomic force microscopy, ExoCAT was observed to have a circular shape with high catalytic activity [100]. Yim et al., developed a photoreceptor-based exosome loading system termed EXPLORs. mCherry was coupled to cryptochrome 2 (CRY2), while CD9 was used to anchor the base helix-loop-helix 1 (CIB1) protein module to the exosomal membrane. Upon blue light irradiation, CRY2 and CIB1 formed a reversible binding, enabling efficient delivery of mCherry to the exosome lumen for subsequent targeting to desired cells [110]. In addition, EXPLORs loaded recombinant luciferase in vitro, albeit with a significantly lower loading capacity than the EXPLOR technique. Furthermore, EXPLORs loaded super suppressor (srlκB) into exosomes and showed promising results in alleviating sepsis in mice [111]. Table 1 outlines these approaches for loading small-molecule, nucleic acid, and protein drugs into exosomes (Table 1).

## 7. Engineered Exosome Targeted Delivery Methods

Despite significant advancements in exosome delivery, effective targeting of engineered exosomes to specific sites such as the brain, liver, lung, and heart for therapeutic purposes remains to be a major research challenge (Figure 3).

### 7.1. Brain Targeting Exosomes

The human brain is protected by the complex and selective blood-brain barrier (BBB), which prevents the entry of most small-molecule drugs and almost all large-molecule therapeutic drugs, thereby, posing a major obstacle to intracerebral drug delivery [112]. However, studies have shown that exosomes derived from various brain cell types (brain endothelial cells [113], microglia [114], astrocytes [115], and choroid plexus epithelial cells [116]) and those originating from metastatic brain tumor [117] can target the brain to varying extents. Designing engineered exosomes with surface-modified targeting peptides represents an effective strategy for achieving brain targeting [118]. For instance, Qu et al., achieved brain-targeted delivery of dopamine by saturating blood exosomes with the drug and subsequently transferring the exosomes into a mouse model of Parkinson’s disease. Since transferrin is abundantly expressed on blood-derived exosomes, these exosomes, via interaction between transferrin and transferrin receptors, transported the drug across the BBB, leading to a greater than 15-fold increase in brain distribution of dopamine and better therapeutic efficacy [50]. Meanwhile, Gao et al., loaded berberine into exosomes by sonication, and found, by in vivo and tissue imaging, that these exosomes exhibited positive targeting towards the brain and spinal cord, potentially improving their anti-inflammatory effect by first crossing the BBB to enter the brain in vivo, and then circulating through cerebrospinal fluid to the site of spinal cord injury [119].

### 7.2. Lung Targeting Exosomes

Lung cancer is a leading cause of cancer-related deaths globally. Precision medicine, including targeted therapy for lung cancer, is currently the most essential direction for treating this disease. Engineered exosomes are a natural vehicle for drug delivery and could be coupled with targeted ligands to achieve specific delivery to lung tissue. For instance, exosomes conjugated with EGFR-targeting peptides and nanoantibodies via protein ligases could facilitate specific uptake by EGFR-positive lung cancer cells. Low doses of engineered exosome-encapsulated PTX have shown significant antitumor effects [120]. In addition to lung cancer, utilizing engineered exosomes has shown potential for the treatment of pulmonary fibrosis. It has been reported that hybridizing CLD-loaded liposomes with fibroblast-derived exosomes could effectively deliver anti-pulmonary fibrosis drugs to tissues. This hybrid system is a promising and specific drug delivery platform for pulmonary fibrosis treatment [64]. Furthermore, exosomes have shown great potential for treating COVID-19-associated lung injury by reducing lung inflammation and promoting the regeneration of damaged alveolar epithelium [121]. Engineering exosomes to target lung tissue and bind to SARS-CoV-2 could prevent and treat lung inflammation caused by viral infection [122].

### 7.3. Cardiac Targeting Exosomes

Exosomes serve as natural carriers for small-molecule, protein, and nucleic acid drugs, making them ideal for targeted delivery to the heart. Extensive studies have been conducted on the use of exosomes derived from MSCs for the treatment of myocardial injury. Improving the targeting ability of exosomes to the heart could advance such studies. Platelet membranes are rich in integrin receptors GPIIb/IIIa and are capable of expressing multiple platelet functions. Transporting platelet membranes to the site of myocardial injury can aid in the treatment of the damaged site [123]. Hu et al., developed platelet membrane hybrid exosomes by mixing platelet membranes with exosomes derived from MSCs and cultured them in control medium containing competing exosomes. The uptake of these hybrid exosomes by endothelial cells was two to three times higher than that of competing exosomes, and their extraction rate by cardiomyocytes was five to eight times higher [124]. Li et al., developed platelet membrane-modified exosomes (P-EVs) via a membrane fusion method. The administered P-EVs were targeted to the ischemic myocardium mediated by monocytes. In the tissue microenvironment, the P-EVs achieved endosomal escape after being phagocytosed by monocyte-differentiated M1 macrophages. The released miRNAs promoted the transformation of M1 macrophages into M2 macrophages, modulating the immune microenvironment and promoting cardiac repair [125].

### 7.4. Hepatic Targeting Exosomes

Exosomes possess intrinsic hepatic accumulation properties. Intravenous injection of exosomes derived from mouse 4T1 mammary tumor cells (4T1-EVs), bone marrow dendritic cells (DC-EVs), and erythrocytes (RBC-EVs) have demonstrated significant accumulation in the liver. It has been confirmed that the accumulation of RBC-EVs in the liver is primarily dependent on C1q-mediated phagocytosis by hepatic macrophages. This natural accumulation property makes RBC-EVs particularly suited for delivering drugs to the liver. Encapsulating drugs within RBC-EVs ensures safety and efficacy for targeted therapy in liver disease [126]. Additionally, neutrophil infiltration is a characteristic pathology of nonalcoholic steatohepatitis. Neutrophil-derived exosomes have been shown to be selectively taken up by the liver, via low-density lipoprotein receptor (LDLR) on hepatocytes and apolipoprotein E (APOE) on the surface of neutrophil-derived exosomes [127]. The liver tissue-targeting properties of hepatocyte-derived exosomes make them suitable carriers for delivering Cas9 RNP, for liver disease-specific gene therapy [98]. For immunotherapy of hepatocellular carcinoma, exosome vaccines have been constructed by anchoring hepatocellular carcinoma-targeting peptides, antigens, and immune adjuvants on the surface of exosomes, inducing effective tumor-specific immune responses via targeted recruitment and activation [128].

## 8. Prospects for Clinical Applications

Exosomes, as highly efficient natural drug carriers, have been extensively explored for the treatment of various diseases due to their low immunogenicity, high stability, and biocompatibility. In recent years, exosomes have played a pivotal role in nanomedicine and precision medicine, becoming promising diagnostic biomarkers and therapeutic tools for diseases, owing to their broad sources and ability to load abundant amounts of specific endogenous cargo. This review primarily focuses on describing recent exosome loading strategies for small-molecule, nucleic acid, and protein drugs. Exosomes are considered to be efficient, cost-effective, and safe drug carriers that are widely used for drug and gene delivery, given their superiority over traditional nanocarriers. In-depth research on drug loading methods could substantially improve therapeutic effects on diseases, since exosomes are classified as active drug components or drug delivery vehicles. Future research on exosomes should primarily concentrate on constructing exosomes to specifically serve drug delivery and clinical efficacy. Existing drug delivery methods suffer from issues of mass production and delivery rates. Thus, researchers should explore alternative drug delivery options in the future. Additionally, the specific functions of exosomes are not yet fully understood, making it challenging to predict their long-term safety and efficacy. Although the modification of targeting peptides on exosomes’ surfaces could substantially enhance their targeting ability, targeting peptides could be potentially immunogenic, leading to possible immune reactions in the human body. Therefore, exploring how to safely and effectively anchor targeting peptides on exosomes is still a problematic issue. It is hoped that more researchers will be involved in resolving these issues in the future. While the practical clinical application of exosomes is still in its infancy, it is believed that exosomes have great prospects for clinical applications as their study and discovery proceed further.

## Figures and Tables

**Figure 1 cells-12-01416-f001:**
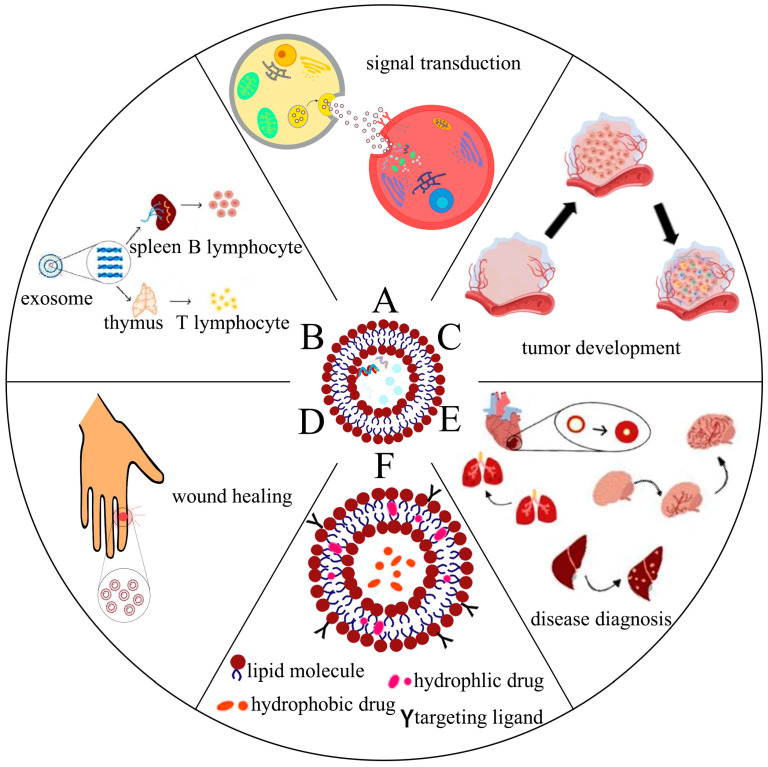
The multiple functions of exosomes: (**A**) Exosomes are involved in intercellular signaling, and cells can exert regulatory effects by secreting exosomes that carry biologically active substances to transmit signals to neighboring or distant cells; (**B**) exosomes can induce immune responses by antigen presentation, regulate immune responses by interacting with immune cells, or produce immunosuppression; (**C**) tumor cell-derived exosomes induce tumor angiogenesis by carrying angiogenesis-stimulating factors such as VEGF, FGF, and IL-8; (**D**) exosomes promote wound healing by regulating macrophages and fibroblasts; (**E**) exosomes are used as biomarkers for disease diagnosis; (**F**) exosomes are natural carriers of drugs, enhancing drug stability and effective drug delivery to the target region.

**Figure 2 cells-12-01416-f002:**
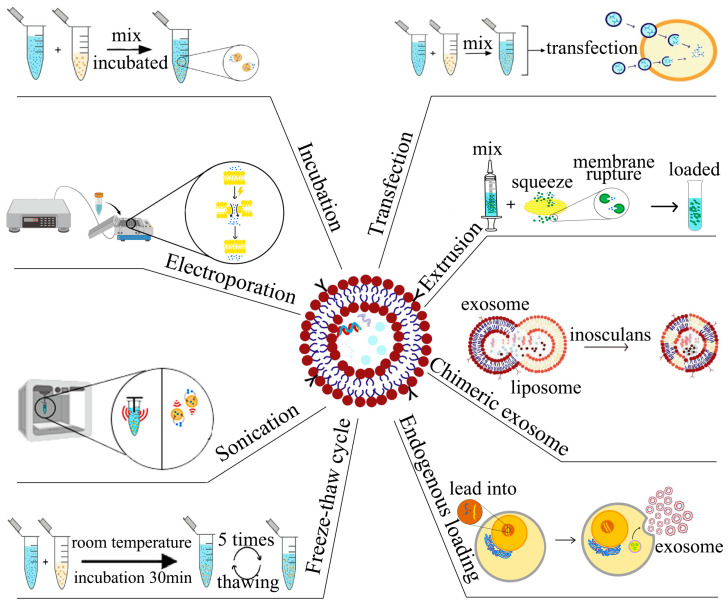
Strategies for exosome drug loading, including incubation, electroporation, sonication, freeze–thaw cycle, transfection, extrusion, chimeric exosome method, and endogenous loading. Different methods need to be performed under different conditions and the drug loading efficiency of exosomes depends on different loading methods.

**Figure 3 cells-12-01416-f003:**
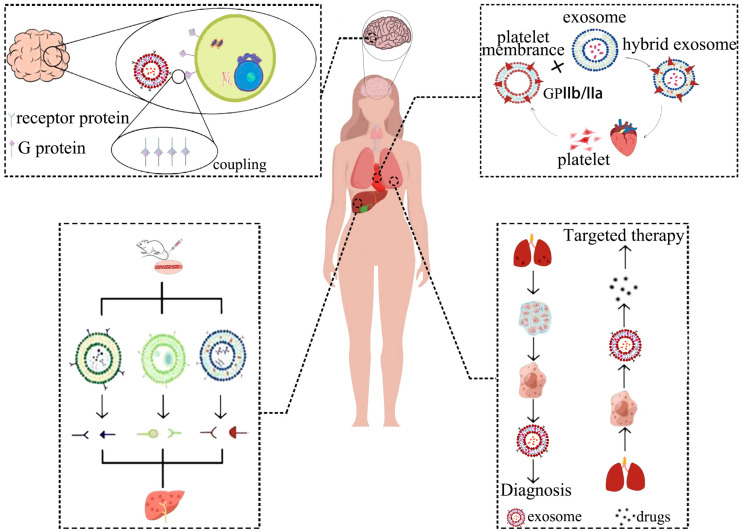
Strategies for targeting exosomes to tissues in vivo. Exosome can penetrate the BBB and enter the brain tissue. At the same time, differently modified exosomes can carry various molecules into organs, such as lung tissue, the cardiovascular system, liver, and other tissues.

**Table 1 cells-12-01416-t001:** Conditions of drug loading, loading efficiency, and exosome source under different drug loading strategies.

Load Strategy	Loaded Drug		Exosome Source	Method for Loading Drug into Exosome	Load Efficiency	Ref.
Incubation	Small molecule	DOX	HEK-293T	4 °C, 2 h	11.73%	[54]
		PTX	Human pancreatic ductal carcinoma cell	Room temperature, 1 h	4.2 ± 0.63%	[55]
		Linezolid	Macrophage	37 °C, 1 h	5.06 ± 0.45%	[56]
	Nucleic acid	miR-159	Macrophage	37 °C, 1.5 h	5.33%	[79]
		siRNA	Mesenchymal stem cell	37 °C, 1 h	43%	[80]
		miR-21-5p	Mesenchymal stem cell	Room temperature, 8 h	1600 copies of miRNA/exosome	[82]
	Protein	catalase	Macrophage	Room temperature, 18 h	4.9 ± 0.5%	[100]
		β-glucuronidase	human mesenchymal stem cells	Room temperature, 10 min	NS	[101]
Electroporation	Small molecule	DOX	Lens epithelial cell	250 V, 350 μF, 4.5 ms	72%	[46]
		HAL	M2 macrophage	100 V, 200 Ω, 100 μF	25.14%	[57]
		Curcumin and SPIONs	Mouse macrophage cell line Raw264.7	400 V, 150 μF, 1 ms	NS	[59]
	Nucleic acid	siRNA	Induced neural stem cell	400 mV, 125 μF, 10–15 ms	17%	[83]
		miR-21 inhibitor oligonucleotides	HEK-293T	1000 V, 10 ms	0.5%	[85]
		miRNA-21	293T	400 V	1.68 ± 0.23%	[86]
		miR-484	HEK-293T	700 V, 150 mF	NS	[87]
	Protein	Insulin	Hepatocellular carcinoma cell	200 V, 50 μF	50.75 ± 1.2%	[102]
			Primary dermal fibroblast		57.42 ± 5.47%	
			Pancreatic β-cell		49.70 ± 4.32%	
		Recombinant Yap1 protein	Platelet	200 V, 500 μF, 26 ms	40.65 ± 3.79%	[103]
		Cas9 RNP	Hepatic stellate cell	NS	20%	[98]
Sonication	Small molecule	PTX	M1 macrophage	20% amplitude, 6 cycles of 30 s on/off for 3 min with a 2 min cooling period between each cycle	19.55 ± 2.48%	[61]
		Curcumin	Macrophage	20% amplitude, six cycles of 30 s on/off for 3 min with a 2 min cooling period between each cycle	0.56 ± 0.01 μg curcumin (1 μg albumin-EVs)	[40]
		Er and RB	HEK-293T	20% amplitude, 6 cycles, 10 s on/off, 3 min duration, 2 min cooling period between each cycle	84% (RB), 60% (Er)	[62]
	Nucleic acid	Bcl-2 siRNA	Milk	30 W, sonicated for 5 s, and then stopped for 2 s for 2 min	66.9 ± 4.5%	[89]
		siRNA-Keap1	Milk	20% amplitude, six 30-s on/off cycles, and a cooling time of 2 min between each cycle	24%	[4]
	Protein	Catalase	Macrophage	500 V, 2 kHz, 20% power, 4 s pulses/2 s pauses, repeat 6 times	26.1 ± 1.2%	[100]
		TPP1	Macrophage	Sonication in water bath at room temperature for 30 min	70 μg TPP1/10^11^ exosomes	[63]
		Cas9 RNP	HEK-293T	2 kHz, 5% power, 2 s pulse/1 s pause and repeated 20 cycles	15.34%	[104]
Extrusion	Small molecule	PTX	huc-MSCs	100 nm pore size extrusion 11 times	14.23 ± 0.25%	[65]
		DA	Induced pluripotent stem cell differentiated from endothelial cell	400 and 200 nm polycarbonate membranes extrusion 10–20 times	45%	[66]
	Protein	Catalase	Macrophage	200 nm pore size	22.2 ± 3.1%	[100]
Freeze–Thaw Cycle	Protein	Catalase	Macrophage	−80 °C fast freeze, thaw at room temperature, repeat cycle 3 times	14.7 ± 1.1%	[100]
		hCG	Uterine fluid	−70 °C fast freeze, thaw at room temperature, repeat cycle 5 times	14.02 ± 5.46%	[106]
		Cas9 RNP	HEK-293T	Freeze in liquid nitrogen, thaw at room temperature, repeat cycle 3 times	37.62%	[104]
Chimeric Exosome Method	Small molecule	TP	Cisplatin-resistant human ovarian cancer cell line	Vortex and sonication (33% amplitude, 2 s pulsed on/off, for 3 min)	78 ± 3%	[71]
		DTX	Fibroblast	Freeze in liquid nitrogen for 5 min, thaw at room temperature for 15 min, repeat 3 times	4.3%	[72]
		CLD	Fibroblast	Vortex and sonication (30% amplitude, 30 s pulse on/off, for 2 min)	70.6 ± 2.7%	[64]
		NIN	Fibroblast	Mixed with EL-CLD in chloroform	90%	[64]
Endogenous loading	Small molecule	PEG-HGNs	Murine melanoma cell	Drug was added to the cells and cultured for 24 h	50%	[74]
		MCC950	Platelet	Incubation at 4 °C for 12 h	7.1%	[75]
Transfection	Nucleic acid	HOTAIR	BDMSCs	pCMV-HOTAIR plasmid, P3000 reagent	Cargo expression increased 974-fold	[90]
		circDYM	HEK-293T	circDYM-GFP lentivirus, Lipofectamine 2000	34.75 copies/exosome	[91]
		miRNA	Human umbilical cord blood-derived mononuclear cell, human urine, foetal bovine serum	Exo-Fect	Above 50%	[92]
	Protein	IFITM3	293T	IFITM3 expression vector, PEI transfection reagent	NS	[107]
		Cas9 protein	HeLa cell, HuH7 cell	HPV-specific and HBV-specific CRISPR/Cas9 expression plasmid	NS	[108]
		CRISPR RNP	Serum	Lipofectamine CRISPRMAX	96%	[109]

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
