# Peer review of "Current Strategies for Exosome Cargo Loading and Targeting Delivery"

_cells, 2023, doi:10.3390/cells12101416_

Round 1

Reviewer 1 Report

This manuscript provides a systematic review of current drug delivery methods in exosomes and summarizes loading strategies of exosomes for small molecule drugs, nucleic acid drugs, and protein drugs.

The manuscript is well-written and provides a comprehensive review. However, the figures are of poor quality (low resolution), difficult to understand and the figure legends need more details to better understand the images.

Author Response

This manuscript provides a systematic review of current drug delivery methods in exosomes and summarizes loading strategies of exosomes for small molecule drugs, nucleic acid drugs, and protein drugs.

The manuscript is well-written and provides a comprehensive review. However, the figures are of poor quality (low resolution), difficult to understand and the figure legends need more details to better understand the images.

Response: Thanks to the reviewer's comments, we have rewritten the details of this manuscript, especially the legends of the figures. The high-resolution figures have been uploaded into the system.

Reviewer 2 Report

Authors reviewed recent studies on different strategy for the loading of cargo in extracellular vesicles. few suggestions need to be included.

Minor comments:

·       Please check the grammar in the manuscript

·       In the abstract exosome introductions is too much. Please rewrite the abstract and try to focus on the applications and problems. Later, why loading of cargo is important.

·       Key words should include Loading of cargo.

·       In the figure1 need to simplify.

·       In figure 2 font are too small to read

·       Figure 3 need to simplify. Things are not visible properly.

Major comments:

·       Review focus on the current strategy of loading of cargo in EVs. In the manuscript last topic is Engineered exosome targeted delivery methods, which is not match with the title. author should rewrite the title of manuscript.

Author Response

Please check the grammar in the manuscript

Response: Thanks for your suggestion. We have revised the grammar in the manuscript.

In the abstract exosome introductions is too much. Please rewrite the abstract and try to focus on the applications and problems. Later, why loading of cargo is important.

Response: Thanks for your good suggestion. We have rewritten the abstract based on this suggestion.

Key words should include Loading of cargo.

 Response: Thanks for your suggestion.

In the figure1 need to simplify.

Response: Thanks for your suggestion.

In figure 2 font are too small to read

 Response: Thanks for your suggestion.

Figure 3 need to simplify. Things are not visible properly.

Response: Thanks for your suggestion.

 Review focus on the current strategy of loading of cargo in EVs. In the manuscript last topic is Engineered exosome targeted delivery methods, which is not match with the title. author should rewrite the title of manuscript.

Response: Thanks for your suggestion. The tile was changed to fit the content better.

Reviewer 3 Report

The article "Current strategies for cargo loading into extracellular vesicles" is up-to-date, detailed, but could be improved:

1)    The title of the paper states "extracellular vesicles", while the abstract and introduction discuss only "Exosomes". Authors should correct this discrepancy - by either narrowing the title or expanding the abstract and introduction of the article.

2)      Correct typos and check for correct spelling of abbreviations, as well as correct spelling of upper/lower case letters in abbreviations, for example: HHighly, aGvHD, c-myc, huc-MSCs, HENPs - this needs to be corrected.

3)    Section 2.1. Extracellular vesicles are capable of delivering organelles in addition to bioactive molecules, for information see doi: 10.3389/fcell.2021.653322

4)    Seems that reference 54 does not match what is written in the text.

5)    The table is not clear, add separators.

6)    Latin names should be written in italics.

7)      Abbreviations written several times: paclitaxel (PTX), docetaxel (DTX)

Author Response

The article "Current strategies for cargo loading into extracellular vesicles" is up-to-date, detailed, but could be improved:

1)    The title of the paper states "extracellular vesicles", while the abstract and introduction discuss only "Exosomes". Authors should correct this discrepancy - by either narrowing the title or expanding the abstract and introduction of the article.

Response: Thanks to the reviewer's comments, we choose to narrow the title.

2)      Correct typos and check for correct spelling of abbreviations, as well as correct spelling of upper/lower case letters in abbreviations, for example: HHighly, aGvHD, c-myc, huc-MSCs, HENPs - this needs to be corrected.

Response: We have checked these abbreviations. We've tried to stay as consistent as possible with the original source.

3)    Section 2.1. Extracellular vesicles are capable of delivering organelles in addition to bioactive molecules, for information see doi: 10.3389/fcell.2021.653322

Response: Thanks for your suggestion. These descriptions were added into the revise manuscript.

4)    Seems that reference 54 does not match what is written in the text.

Response: Thanks. We have revised this mistake.

5)    The table is not clear, add separators.

Response: Thanks for your suggestion.

6)    Latin names should be written in italics.

Response: Thanks for your suggestion. We have revised this issue.

7)      Abbreviations written several times: paclitaxel (PTX), docetaxel (DTX)

Response: Thanks for your suggestion. We have unified the abbreviations throughout the text.

Round 2

Reviewer 2 Report

Author included all the queries and suggestions in revised manuscript.